# CNN Variational Autoencoders' Reconstruction Ability of Long ECG Signals

## Abstract

Can variational auto-encoders (VAEs) generate flexible continuous latent space for long electrocardiogram (ECG) segments and reconstruct the input? A folded VAE architecture is introduced in this study which is able to encode long ECG segments by splitting an input segment into folds and process them in sequence using a narrow field-of-view in the encoder and concatenate them at the end, instead of processing the long segment at a time. The VAE decoder follows similar folding and concatenation strategy for reconstruction of the original ECG segments. The proposed folded VAE architecture is able to generate better reconstruction of long 30-second ECG segments compared to unfolded classical VAE approach which often produce trivial reconstruction of long ECG segments. Experimental results show that the latent representation generated by our folded VAE architecture not only retains rich compressed information but also aids designing interpretable models by providing decision-making insights.

## 1 Introduction

Variational autoencoders (VAE) are powerful tool for generative modeling. Due to their inherent continuous latent space, operations such as random sampling and interpolation becomes possible. The reconstruction ability of VAEs is an important feature which has many applications including prototype-based deep learning interpretability methods for time-series (Gee et al., 2019; Zhang et al., 2023; Li et al., 2018). Reconstruction of VAEs for longer physiological time-series, such as electrocardiogram (ECG), of various morphology and signal quality remains under explored.

Prototype classifiers are VAE networks where observations are classified based on their similarity to one or more prototype observations within the dataset (Li et al., 2018). The proximity between prototype and observations is measured in a flexible and adaptive latent space (Li et al., 2018; Gee et al., 2019; Zhang et al., 2023). Reconstruction ability of VAE's decoder module of prototypes and observations is crucial for communicating interpretability to the end users, as well as, to validate the claimed prototypes against a given observation for knowledge exploration. Reconstruction of ECG data using VAEs has been explored in several occasions in the literature including prototype learning in intrinsic interpretability literature of deep learning (Gee et al., 2019; Zhang et al., 2023), and ECG generation using VAEs (Kuznetsov et al., 2021; Beetz et al., 2022; Jang et al., 2021). ECG segment reconstruction, which was often attempted in these studies, either produce short 3-10 second ECG segments or healthy heart beats such as sinus rhythms. The quality of VAE-sampled latent representation and the ability of VAE's decoder module to reproduce the source sample needs to be investigated for long ECG segments and a broad spectrum of heart beats.

VAEs well reconstruct shorter ECG segments, as shown in Figure 1-a, but generates trivial reconstruction for longer 10 seconds and 30 seconds fragments, shown in Figure 1-b,c. This limitation of VAEs' reconstruction ability of ECG data needs to be improved to be useful for respective applications such as visualisation of prototype and observation in deep learning interpretability studies based on prototype learning.

For improving the reconstruction ability of VAEs, motivated by the concept of manifold learning which considers that the dimensionality of many data sets is only artificially high, we propose a folded encoder/decoder architecture of VAE which narrows down the input field-of-view by splitting a long ECG input segment into small folds at the beginning of the encoder/decoder followed by a

concatenation at the end. The consumption of a long segment into folds by the encoder/decoder backbone allows the network to learn essential ECG features and produce better reconstruction.

## 2 METHODOLOGY

### 2.1 PROBLEM FORMULATION

The problem of limited reconstruction ability of VAEs of long ECG segments (shown in Figure 1 bottom-row) is formulated to stimulate the latent space representation, generated by VAEs, to be able to capture pattern in relatively longer ECG sequences for better reconstruction of the input sequence. The encoder/decoder network architecture of CNN variational auto-encoders were experimented for suitability in learning latent space representation.

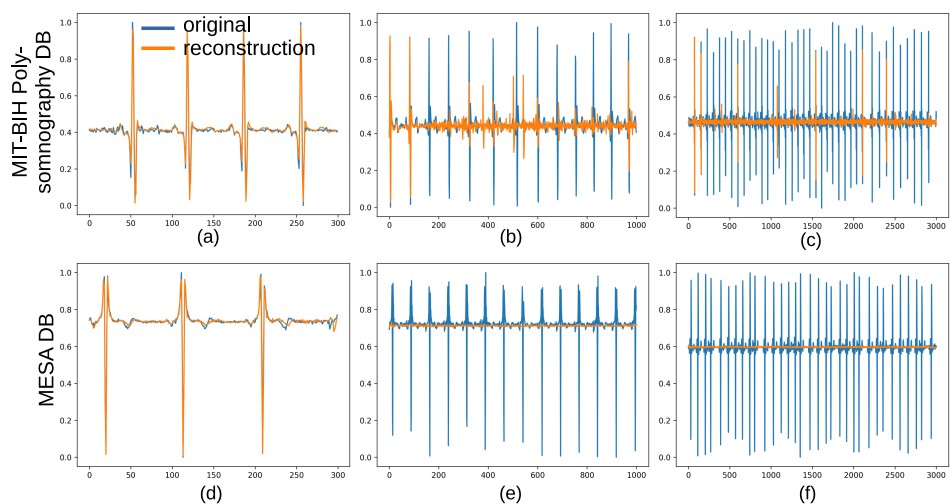

Figure 1: Original (blue color) and reconstructed (orange color) ECG signal fragment for CNN variational auto-encoder of (left column: a, d) 3 second, (mid column: b, e) 10 second and (right column: c, f) 30 second fragment unsupervised trained on ECG recordings of MIT-BIH Polysomnographic Database (top row) and Mesa dataset (bottom row). The reconstruction ability of VAEs suffers due to increase in segment length (3, 10 and 30 seconds) and variation in ECG signal recording environment (above-mentioned two databases).

### 2.2 VARIATIONAL AUTO-ENCODERS (VAEs)

Autoencoders (AEs) are one kind of neural network which consists of two parts - encoder $e$ and decoder $d$ (Figure 2-a, excluding the *sampler* which can sample from latent distribution). The *encoder* produces compressed latent representation $Z$ from original high-dimensional input space $X$ by learning a non-linear transformation $e : X \rightarrow Z$. The *decoder*, on the other hand, learns a transformation $d : Z \rightarrow X$ which tries to recover $X$ from the latent representation $Z$. AEs map the input to a latent vector in a deterministic way $z = e(x)$ which limits the decoders to reconstruct $\hat{x} = d(z) = d(e(x))$ from a region in the latent space that was never seen by the decoder during the training (Ghosh et al., 2019).

VAEs, on the other hand, sample a latent vector $z \sim N(\mu, \sigma)$ by parameterizing a diagonal Guassian distribution $N(\mu, \sigma)$ where the parameters $\mu(x)$ and $\sigma(x)$ are mapped from input $x$ (Kingma & Welling, 2013).

## 2.3 FOLDING AN ECG SIGNAL

Motivated by the principle of *manifold* learning, which states the idea that the dimensionality of many data sets is only artificially high (Martinetz & Schulten, 1994; Roweis & Saul, 2000), the problem of encoding an $N$ second ECG recording should be broken down to encoding $N$ number of only one-second ECG segments, which is likely to contain as few as one or two heart beats, followed by a concatenation of encoded folded segments to yield the final encoding. A folded encoding and decoding can be summarised by the Equation 1 and 2,

$$e(x) = \sum_{i=1}^{n} e(x_i) \tag{1}$$

$$\hat{x} = d(z) = \sum_{i=1}^{n} d(z_i) \tag{2}$$

where $e(x)$ and $d(z)$ are encoding and decoding which are achieved by applying them over folded inputs $x_i$ and folded sampled vector $z_i$ respectively.

Figure 2-a, b show the concept of ECG folding, in encoder - concatenating encoded segments, sampling from the merged encoding and handing to the decoder, where the opposite is performed including folding the sampled encoding, decoding each fold and finally merging them to yield the final reconstruction. A single network backbone was used by the encoder to encode each fold and the similar shared backbone strategy was followed in the decoder. The use of a shared encoding (or decoding) sub-network forces it to learn distribution of ECG beats (such as, folds in a manifold ECG recording) to encode (or decode). The use of a shared sub-network is expected to enforce in learning general ECG heartbeat features discarding unnecessary information including noise.

## 2.4 VAE ARCHITECTURE

Figure 2-a shows the concept of shared backbone encoder and decoder for folded ECG and Figure 2-b shows a block diagram of corresponding neural network layers for 30 second long ECG input sampled at 64Hz.

The encoder in Figure 2-b consists of four convolution-blocks which receives 30 one-second ECG folds (1920 samples of a 30 second segment sampled at 64Hz turns into 30 64-sample folds) in sequence and outputs a feature-map consisting of 4 spatial samples. A fixed number of 8 output channels per convolution layer was used to keep the network complexity low (thus, the encoder outputs a 8x4 feature-map). A concatenation layer merges 30 8x4 feature-maps in the second dimension (spatial dimension in channel $X$ spatial space) yielding a 8x120 feature-map which moves to the sampling layer. The use of pooling layer was avoided to retain the temporal relationship from bottom to top of the four convolution layers and a convolution stride of 2 was used to down-sample the input similar to a pooling layer.

The sampler takes the merged feature-map of 8x120, flattens to 1x960 vector and maps to mean vector $\mu(x)$ and standard deviation vector $\sigma(x)$ of a reduced dimension 1x480 which forms the basis of a Guassian distribution $N(\mu, \sigma)$ from which a latent vector $z$ was sampled and passed to the decoder.

The decoder receives a 1x480 latent vector $z$, maps it back to 1x960 and reshapes to 8x120 which is the dimension same as the concatenated output of the encoder. A folded strategy was adopted, same as the encoder, by splitting the 8x120 feature-map into 30 8x4 feature-maps which goes through four transposed convolution blocks, each producing a 1x64 output feature-map. A concatenation layer then merges 30 of these feature-maps and produce a final 1x1920 feature-map which is the same shape as the original input. The transposed convolution block in the decoder consists of an up-sampling layer of scale 2 and a convolution layer with stride 1. A leaky-ReLU non-linearity activation was used in both encoder and decoder to avoid the dying ReLU problem of large number of zero neurons due to negative bias (Maas et al., 2013).

## 2.5 VAE UNSUPERVISED TRAINING

The VAE was trained unsupervised on two different ECG segment lengths - 10 second and 30 second with 10-folding and 30-folding respectively to verify if VAE reconstruction of per-second folding

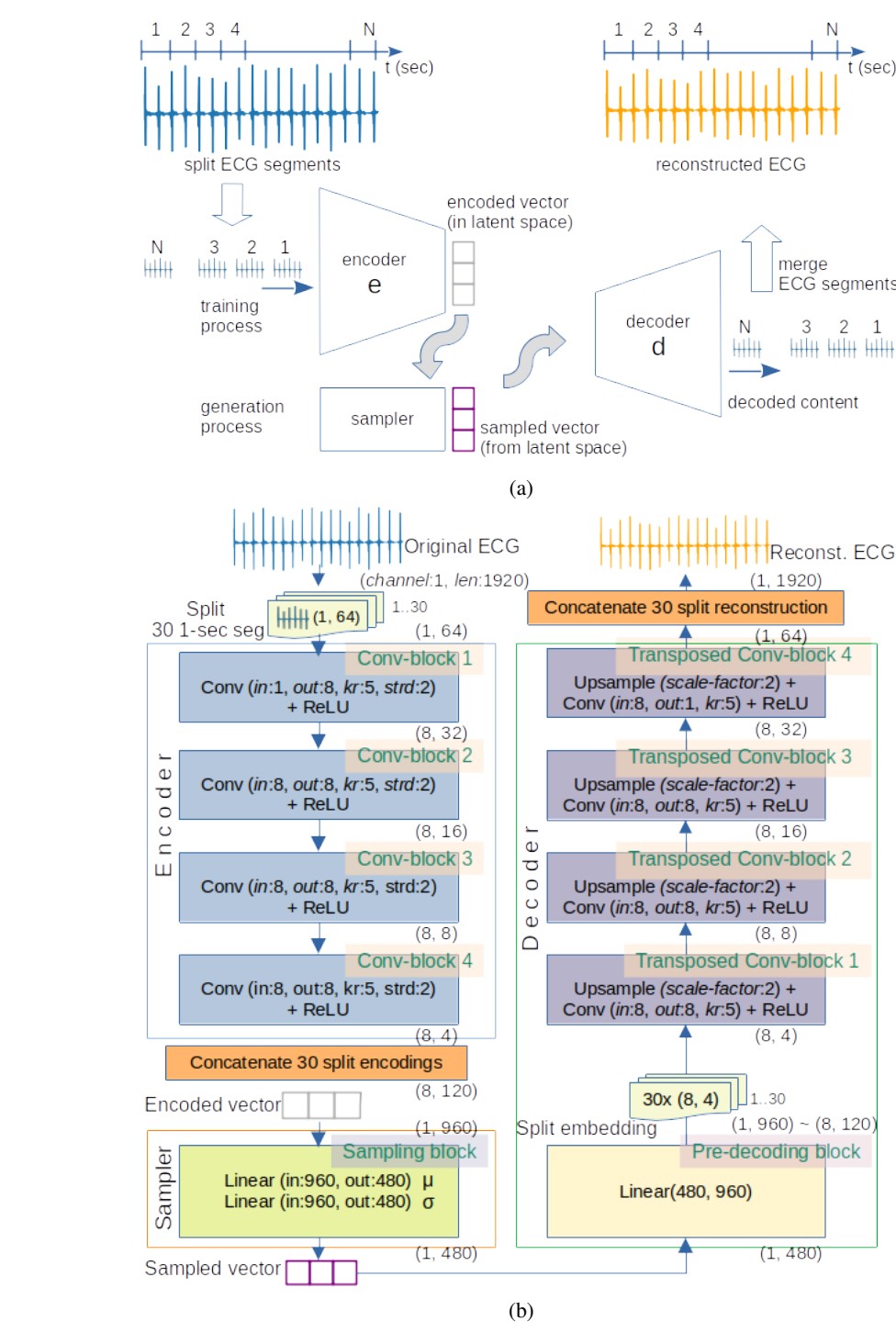

Figure 2: (a) CNN VAE concept diagram and (b) architecture with shared feature-extracting backbone. ECG N second segment is split into N sub-segments of 1 second excerpts, then encoded, sampled and forwarded to decoder for reconstructing sub-segments which are then merged to output N second segment the same length as the original ECG.

yields similar results. The objective of the VAE network is to minimise a loss $L_{VAE}$

$$L_{VAE} = w_f * L_{recon} + L_{KL} \tag{3}$$

which consists of 2 parts - reconstruction loss $L_{recon}$ and KL Divergence loss $L_{KL}$. The reconstruction loss ensures that the ECG segment generated by the decoder closely resemble the input ECG and for this, binary cross-entropy (BCE) loss was found to be convenient given that the input ECG was scaled using minmax scalar (0-1 range) and decoder outputs pass through a *sigmoid* layer to predict output ECG in 0-1 range. BCE loss for a batch of $n$ samples can be formulated as below -

$$L_{recon}(x, \hat{x}) = -\omega_n[\hat{x}_n.logx_n + (1 - \hat{x}_n).log(1 - x_n)] \tag{4}$$

where $x$ and $\hat{x}$ are original and predicted ECG segments accordingly. The reconstruction loss $L_{recon}$ in the equation of $L_{VAE}$ (Equation 3) was scaled by a factor of $w_f$ to balance the reconstruction and KL loss components which corresponds to the fidelity of the reconstructions and the regularity of the latent space. A value of 500 was found suitable for $w_f$ in this experiment.

The KL Divergence loss $L_{KL}$ can be stated as

$$L_{KL}[G(Z_\mu, Z_\sigma)|N(0, 1)] = -0.5 * \sum_{i=1}^{N} 1 + log(Z_{\sigma_i^2}) - Z_{\mu_i}^2 - Z_{\sigma_i}^2 \tag{5}$$

where $G(Z_\mu, Z_\sigma)$ is the Guassian distribution defined by the encoder's output $Z_\mu$ (mean) and $Z_\sigma$ (standard deviation), and $N(0, 1)$ is the standard normal distribution. The KL Divergence loss $L_{KL}$ measures the difference between the encoder's distribution and a prior which is a standard normal distribution in this experiment. It acts like a regulariser which ensures the latent variables are close to a prior distribution.

## 2.6 USE CASE: SLEEP STAGE CLASSIFICATION FROM ECG

In addition to verifying VAE reconstruction ability using shared encoder/decoder backbone and second-wise folded ECG, the quality of the sampled latent space representation was also tested by using the representation in a context of 3-stage sleep classification from ECG signals including Wake, NREM- and REM-sleep. It was hypothesized that the performance of folded ECG with shared VAE encoder/decoder backbone should not perform lesser than the performance of an unfolded standard VAE scenario.

## 2.7 DATASET

Multi-Ethnic Study of Atherosclerosis (MESA) (Zhang et al., 2018a; Chen et al., 2015) dataset was used for the use case of sleep-wake classification. The dataset itself was sourced from the National Sleep Research Resource which provides availability of the de-identified records (Dean et al., 2016; Zhang et al., 2018b). Due to a large number of recordings, a randomly selected subset of 20 subjects was chosen for the current study. ECG recordings in the dataset were sampled at 256Hz which were downsampled to 100Hz. Another dataset was MIT-BIH Polysomnographic Database (SLPDB) (Ichimaru & Moody, 1999; Goldberger et al., 2000) from Physionet repository which contains 18 recordings of multiple physiological signals including ECG during sleep. Sleep stages were annotated as 30 second segments. Original 250Hz recordings were downsampled to 100Hz.

## 2.8 NETWORK ARCHITECTURE

The high-level hybrid explainable architecture is shown in Figure 3. The architecture adds two additional components to the VAE module described above: The Parameterizer and Aggregator.

The Parameterizer is inspired by Alvarez-Melis & Jaakkola (2018); Zhang et al. (2022) and is used to generate relevance weights of the different splits for each input sample. Whereas in Zhang et al. (2022) the Parameterizer module was used to generate relevence weights for encoded concepts generated from a self-attention mechanism, here the Parameterizer aims to do the same for the predefined splits. It is important to note that unlike the encoder of the VAE, the Parameterizer accepts the entire signal. This provides the Parameterizer with a higher level of context across the entire 30 second sample without the VAE being required to compute over the entire signal. The output of the Parameterizer is a vector $\boldsymbol{w} \in \mathbb{R}^N$ where $N$ is the number of splits and $w_n \in \boldsymbol{w}$ is the weight for split $n$, for a given sample.

The Aggregator first calculates the linear combination of the flattened encoded splits (or folds) with the relevance weights and produces a probability weighted by the predicted relevance of each split.

The encoded splits are flattened using a fully connected layer where the output is a vector $\boldsymbol{p} \in \mathbb{R}^C$ where $p_c$ is the probability of the entire sample being class $c$.

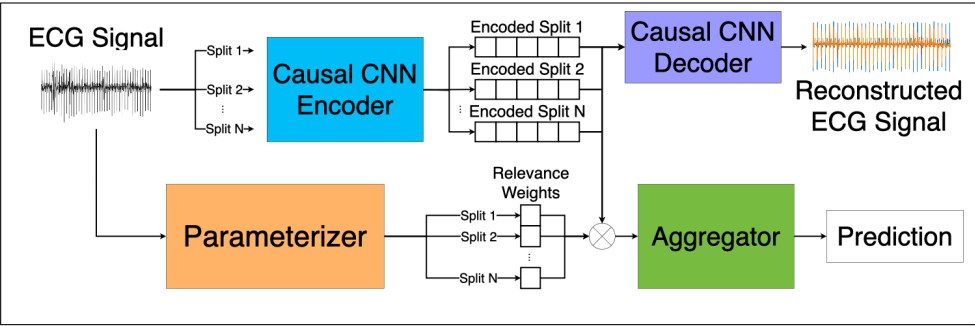

Figure 3: High-level explainable architecture with segmentation and parameterizer.

While the VAE portions of the model remained fixed as described in section 2.4, Table 1 describes the output shape, kernel size, and number of parameters for the enconder of the Parameterizer module.

| Layer Type | Output Shape | Kernel Shape | Number of Parameters |
|---|---|---|---|
| Conv1d | [32, 32, 2996] | [5] | 192 |
| BatchNorm1d | [32, 32, 2996] | – | 64 |
| Conv1d | [32, 32, 2996] | [5] | 5,152 |
| BatchNorm1d | [32, 32, 2996] | – | 64 |
| Swish | [32, 32, 2996] | – | – |
| Conv1d | [32, 32, 2996] | [5] | 5,152 |
| BatchNorm1d | [32, 32, 2996] | – | 64 |
| Swish | [32, 32, 2996] | – | – |
| MaxPool1d | [32, 32, 1500] | 2 | – |
| AdaptiveMaxPool1d | [32, 32, 1] | – | – |
| Trainable params: 11,082 | | | |
| Non-trainable params: 0 | | | |

Table 1: Parameterizer Model Architecture

## 2.9 MODEL TRAINING AND TESTING

The classifier sub-network is trained simultaneously with the VAE and the objective of the network is to minimise the combined reconstruction and classification loss,

$$L = L_c(X_i, y_i) + L_{VAE} \tag{6}$$

Cross-entropy loss was found suitable to account for the classification error and the combined loss was back-propagated through the whole network (classifier and VAE encoder/decoder sub-networks) by using ADAM optimiser for faster convergence. A small learning rate 0.0001 was found suitable for training the network with maximum epoch as high as 400. A validation loss was calculated end of each epoch and any improvement in validation loss was considered as a milestone throughout the end of the maximum epoch and during each milestone the model was saved and at the end test performance was calculated with the latest model.

Subject-wise split was considered for the sleep-stage classification task in both the datasets where a 80/20 split of subjects was done for training and testing and the training subjects were further split into 80/20 for training and validation. Recordings in training were then segmented into 30 second sleep epochs, mixed together and used for model training and a similar strategy was adopted for the validation and testing groups. Pre-processing steps include downsampling to a common 100Hz frequency and removal of baseline wander in ECG signals.

## 2.10 EXPERIMENTAL SETUP

There were 2 experiments carried out in this study. Firstly, a folded VAE architecture was used to train unsupervised way from both the datasets - MESA dataset and MIT-BIH Polysomnography dataset. This one is to validate if the problem specified in Figure 1 can be solved in both the datasets. Secondly, a VAE guided network arrangement was used to train a sleep stage classifier to learn 3 stages such as Wake, NREM- and REM-sleep stages. Particularly, 6-fold and 10-fold VAEs were used for the VAE guided sleep-stage classification task. A note on the Figure 2 is that the 64Hz sampling frequency of input ECG shown in the network diagram was altered in the experiment to be 100Hz for convenience and it is suggested to consider the network diagram as a representative model.

## 3 RESULTS

Figure 4 shows the reconstruction of 30 second ECG segments with 3, 5 and 10 splits before passing through the encoder/decoder and finally concatenate at the decoder output. The reconstruction was

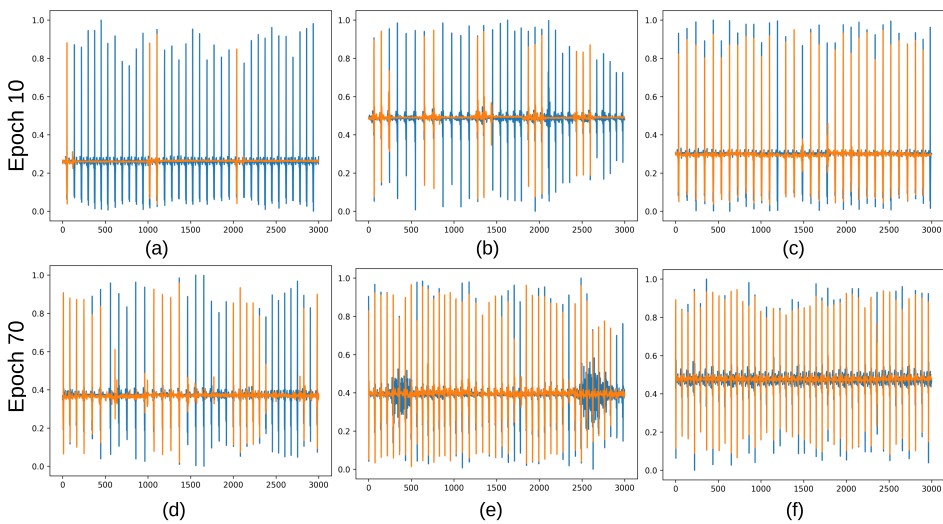

Figure 4: Reconstruction of CNN VAE with shared feature-extracting backbone which splits input 30 second segment into (left column: a, d) 3-folds, 10-second each, (middle column: b, e) 5-folds, 6-second each and (right column: c, f) 10-folds, 3-second each. The top row shows reconstruction at 10 epoch of Split-VAE training and the bottom row shows that at 70 epoch. The ECG data are from MIT-BIH Polysomnographic Database.

captured during unsupervised training of VAE using the Physionet Polysomnographic dataset at 10th (Figure 4:a-c) and 70th (Figure 4:d-f) training iterations. Reconstructed ECG during early training iterations (epoch 10) of both 10 second and 30 second segments were found to be sparse than the reconstruction during later iteration (epoch 70). Reconstructed 3-split and 5-split of 30 second ECG segments (Figure 4:{a, d} and {b, e}) were found to recover fewer ECG beats compared to 10-split scenario (Figure 4-c, f).

Figure 5 shows reconstruction of 30 second segments of 20 randomly selected ECG recordings in 6- and 10-splits after 60 epoch of VAE training. The reconstruction quality shows that the decoder can recognise every ECG beat of 30 second segments.

The sleep stage classification accuracy across all four test subjects of the MESA dataset was 71.64%, 75.17%, 69.03%, and 44.15%, with an mean accuracy of 65.00%. For 6-split VAE, the Parameterizer generates weights as $[\mathbf{0.774}, 0.688, 0.693, 0.693, 0.696, 0.692]$ for one of the test subjects which signifies the contribution of each of the 6-folds. The first weight $0.774$ out stands which demands a closer inspection of the first 5 seconds of the input ECG to understand any pattern and do domain specific interpretation. Likewise, for 10-split VAE, the Parameterizer generates weights

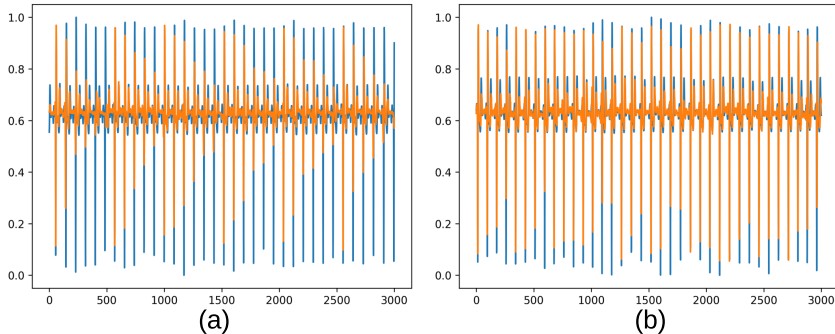

Figure 5: Reconstruction of CNN VAE with shared feature-extracting backbone which splits input 30 second segment into (a) 6 folds, and (b) 10 folds at epoch 70 of VAE training. The ECG data are from 20 randomly selected subjects of MESA Database.

as $[\mathbf{0.939}, 0.892, 0.860, 0.732, 0.854, 0.799, \mathbf{0.939}, 0.635, 0.816, 0.730]$ for one of the test subjects which signifies the importance of the highlighted folds (fold number 1 and 7).

## 4 DISCUSSION

The results from unsupervised of the VAE architecture indicate a good pattern of reconstruction for a higher number of splits. This aligns closely with the reconstruction capabilities demonstrated in previous VAE studies on ECG which demonstrated good reconstruction for 3-10 second samples (Kuznetsov et al., 2021; Beetz et al., 2022; Jang et al., 2021). The poorer reconstruction for 3 and 10 fold variants provide further evidence that CNN-based VAE architectures struggle to encode long ECG signals with sufficient granularity as discussed in Figure 1.

Interestingly, though reconstruction was improved by the addition of more data splits, the classification performance of the resultant model was poor compared to other architectures trained on the MESA dataset. For example, Erdenebayar et al. (2020) used a deep network of gated-recurrent units for three-class sleep stage classification achieving an 80.07% test accuracy while Tang et al. (2022) utilised a CNN feature extractor combined with a novel transfer-learning approach to achieve high generalisability for sleep stage classification across subjects with an average accuracy of 80.6%. Previously, Immaculate Joy et al. (2024) demonstrated a high level of accuracy for cardio-vascular disease detection through a similar method of using the latent space encoding of a VAE model as the learned features for a classifier. This discrepancy suggests two possible conjectures for why our current model was unable to perform. Firstly, the model's hyperparameters may not be correctly optimised, leading to poor generalisability to the test set. Indeed during training, a level of over-fitting comparing the loss on training and validation in later epochs. Combined with the good reconstruction, this may indicate that the Parameterizer module may be too large. Another conjecture for the model's poor performance may be that the information between splits is not being captured by the VAE encoder and subsequently not being adequately reflected in the relevance weights generated by the Parameterizer when encoding the entire signal. The American Sleep Academy of Sleep Medicine (AASM) defines features from multiple heartbeats as relevant to sleep stage classification as patterns such as sustained bradycardia or tachycardia, which provides some biological precedence for inter-split information (Iber et al., 2007).

The folded-VAE architecture, equipped with Parameterizer and Aggregator modules, provides a flexible framework to design an intrinsically interpretable deep learning model architecture similar to (Alvarez-Melis & Jaakkola, 2018; Zhang et al., 2023). The only exception is (Zhang et al., 2023) was trying to connect the neighboring sample points to form concepts, while in our folded VAE strategy, each fold is seen to be a concept as granular as required based on the given problem. For example, for a single heart beat level variation, a 30 second long ECG segment can be divided into 30-splits which would be a single second-wise split VAE where the encoder would try to learn a single beat on average in a single second ECG segment. Sleep stage classification problem probably require to observe heart-beat pattern from a number of consecutive beats which can be considered

as a hyperparameter; we have explored 6-fold and 10-fold only in current study and requires further investigation.

The sampling step in the VAE encoder used the concatenated folds to sample the encoding which then acts as an input to the decoder (shown in Figure 2). It can be argued that the sampler may find it difficult to merge information between folds to form a continuous space. An alternative strategy can be to take sampling each fold and then concatenate the sampled folds. This sampling-folds first, followed by concatenation to form VAE encoder output is used in generating the reconstructions shown in the results (Figure 4 and Figure 5) with no phase shift issue of the folds in the reconstruction. Any further study should consider this alteration of the sampler and concatenation into account.

## 5 CONCLUSION AND FUTURE WORK

Variational autoencoders are growing in popularity for ECG analysis because they allow for simple embedding into a latent space that can be easily sampled from. However, most studies focus on applications where the duration of ECG signal being analysed is short. The current study aimed to investigate the capability of CNN variational autoencoders on long ECG signals and suggest a potential architecture to address shortcomings by splitting the ECG into smaller chunks. Though good reconstruction was observed for a higher number of splits, classification accuracy for test subjects is lower than other architectures trained on the same dataset. Future work is required to determine the exact mechanism for the poor classification accuracy, and to improve on the initial interpretable architecture proposed.

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
