# OpenReview forum: "CNN Variational autoencoders' reconstruction ability of long ECG signals"
_ICLR.cc/2025/Conference — Submitted to ICLR 2025_

### Official Review · Reviewer_X6ep · 2024-10-22

**Soundness:** 2
**Presentation:** 2
**Contribution:** 2
**Rating:** 3
**Confidence:** 4

**Summary:**

This work explores the use of a folded Variational Autoencoder (VAE) architecture to reconstruct long electrocardiogram (ECG) signals. A folded VAE architecture is proposed to address the limitations of traditional VAEs in handling long ECG sequences. The proposed method involves splitting long ECG segments into smaller folds, processing them sequentially, and concatenating them for reconstruction, within a VAE framework. The authors evaluate the proposed method's performance in a sleep stage classification task using two datasets.

**Strengths:**

The question of whether variational auto-encoders can generate a flexible continuous latent space for long electrocardiogram (ECG) segments and reconstruct the input is interesting.

**Weaknesses:**

-	Related work section is missing completely. Can the authors comment on what work is out there that investigated similar problems as they do? How do for example the following works relate to the work proposed here (just to name a few):
1. Comparison of Autoencoder Encodings for ECG Representation in Downstream Prediction Tasks. Christopher J. Harvey, Sumaiya Shomaji, Zijun Yao, Member, IEEE, Amit Noheria , 2024
2. Multi-Domain Variational Autoencoders for Combined Modeling of MRI-Based Biventricular Anatomy and ECG-Based Cardiac Electrophysiology , Marcel Beetz, Abhirup Banerjee and Vicente Grau, Frontiers in Physiology, 2022
3. Joint optimization of a β-VAE for ECG task-specific feature extraction. Viktor van der Valk, Douwe Atsma, Roderick Scherptong, and Marius Staring, arXiv 2023
4. Feasibility of ECG Reconstruction From Minimal Lead Sets Using Convolutional Neural Networks, Maksymilian Matyschik; Henry Mauranen; Pietro Bonizzi; Joël Karel, IEEE 2020 Computing in Cardiology


As no related work is mentioned, also no baselines or comparisons are performed. I think the simplest comparison would be to take the vanilla VAE and take short sequences, and then concatenate. How does this compare to the proposed folded VAE?

Further the presentation of results is very poor. There are long vectors of numbers put into the text. Please create e.g. tables that show what method you use, what the result is for different architectures or modifications, and display this in a structured way.

**Questions:**

see above

---

### Official Review · Reviewer_YF1s · 2024-10-23

**Soundness:** 2
**Presentation:** 2
**Contribution:** 2
**Rating:** 1
**Confidence:** 5

**Summary:**

This paper presents a VAE for reconstructing long (10-30s) ECG sequences. This is achieved with a so-called folding scheme to process short sequences during the encoding and then combine the folds before a decoder mirroring a similar architecture. Experiments are conducted on sleep ECG from two different datasets, and the benefit of the approach is investigated in both reconstruction and its effect on downstream sleep classification.

**Strengths:**

1. The problem of reconstructing long ECG sequence is interesting.

2. The experiments were conducted on 2 datasets and consider not only reconstruction but also classification tasks for performance evaluation.

**Weaknesses:**

1. The premise of the study can be made more clear. Currently It is not clear what is the benefit of the presented method against simply using a sliding window on a long ECG sequence by reconstructing short sequences across the windows

2. Some of the technical components can be better described. For instance, it is not clear what the summation means in equations (1-2) — does it really mean summation by averaging over the representation obtained from different folds? If yes, why? Similarly, it was not well justified why we can merge 30 8x4 features into 8x120 future map.

3. The relation between the VAE described in 2.4 and the specific model described in 2.8 is not clear.

4. There were baselines or comparative studies conducted evaluating the presented method with existing approaches to handle this, especially by simply learning to reconstruct for short sequences and apply to long sequences with a sliding window.

**Questions:**

1. To more clearly  demonstrate the advantages of the presented method, please explicitly compare the folding approach to a sliding window baseline, both conceptually and empirically.

2. Please clarify the questions raised in bullet 2 in the weakness section. Furthermore, please provide a step-by-step explanation of how the folding and merging operations are implemented, including the mathematical justification for each step.

3. Please provide a diagram or explicit description of how the VAE from section 2.4 is integrated into the overall architecture described in section 2.8. This would help clarify the relationship between these components.

---

### Official Review · Reviewer_799m · 2024-10-30

**Soundness:** 2
**Presentation:** 2
**Contribution:** 1
**Rating:** 3
**Confidence:** 4

**Summary:**

This paper introduces a folded VAE architecture designed to improve the reconstruction quality of long ECG segments, aiming to retain interpretable features for downstream tasks like sleep stage classification. The authors present a CNN-based VAE architecture that encodes ECG signals by folding long segments into smaller sub-segments and processing them through a shared backbone encoder and decoder. The manuscript explores this architecture's reconstruction quality, highlights the challenges of using VAEs for long physiological signals, and evaluates the model's performance on two ECG datasets. Additionally, the folded-VAE framework's latent space representation is leveraged for sleep stage classification.

**Strengths:**

- Comprehensive Experimentation: The paper includes experiments with multiple datasets (MESA and MIT-BIH), demonstrating the generalizability of the proposed architecture across different signal sources and classification contexts.
- Practical Applications: The application of this model to sleep stage classification is valuable and opens doors for future research on ECG-based sleep monitoring systems, which has relevance in healthcare.

**Weaknesses:**

- Lack of Quantitative Metrics for Reconstruction Quality: The reconstruction results rely on visual analysis without quantitative metrics, which may lead to subjective conclusions.
- Suboptimal Sleep Stage Classification Results: The model’s classification accuracy (mean of 65% across subjects) is lower than other baseline methods.
- Over-Reliance on Folding Technique Without Comparison to Alternative Approaches: While folding offers improved reconstruction, the authors do not compare this method against alternative architectures (e.g., hierarchical or multi-scale VAEs, or LSTM-based methods) that could also encode long signals effectively.
- Clarity and Language: Some sections, particularly in the methodology, could benefit from improved clarity. For instance, equations used to describe the encoding and decoding of ECG segments could be elaborated on to better illustrate the folding approach.

**Questions:**

# Suggestions

- Lack of Quantitative Metrics for Reconstruction Quality:  --> Idea: Including quantitative reconstruction metrics (e.g., Mean Squared Error, Structural Similarity Index) would make it easier to assess the improvement over standard VAE architectures objectively.
- Suboptimal Sleep Stage Classification Results:  --> Idea: Exploring methods to enhance inter-segment information sharing could improve performance. Incorporating recurrent layers (RNN, LSTM) in the Parameterizer module could capture temporal dependencies between ECG segments, potentially boosting classification accuracy.
- Over-Reliance on Folding Technique Without Comparison to Alternative Approaches:  --> Idea: Including a baseline comparison with alternative architectures for long-sequence encoding would provide a clearer picture of the folding method’s relative effectiveness.

---

### Official Review · Reviewer_RHvA · 2024-11-06

**Soundness:** 1
**Presentation:** 1
**Contribution:** 1
**Rating:** 1
**Confidence:** 4

**Summary:**

The paper aims to build a VAE-based model for long ECG segments and considers a benchmark ECG dataset for the analysis.

**Strengths:**

The paper's application area is an important problem of long-sequence reconstruction, which could enable the capture of essential clinical information in the latent space.

**Weaknesses:**

1. The paper's presentation is poor and does not meet the standards of ICLR or other relevant AI/ML venues. The authors are advised to proofread the paper carefully, as there are numerous grammatical errors, including missing commas, throughout the text, including in the abstract. Many of these errors could be easily fixed by using available grammar checkers, suggesting that the manuscript may not yet be ready for submission.
2. The proposed method lacks novelty, as it primarily focuses on splitting signals in the input space.
3. The results are not well-presented, making it unclear what the main contributions of this work are.

**Questions:**

N/A

---

### Meta-Review · Area_Chair_prq4 · 2024-12-15

**Metareview:**

This paper introduces a folded VAE architecture which is able to encode long EEG segments, aimed at generating better reconstruction of long EEG segments. Experiments on two ECG datasets are performed to verify the performance of the proposed model.  All of reviewers agree that the current paper is not well prepared, so that the presentation should be dramatically improved.  Related work is completely missing, so read the reviewers' comments carefully to include them in future submissions. It was pointed out that it is not clear what is the benefit of the presented method against simply using a sliding window on a long ECG sequence by reconstructing short sequences across the windows.
Therefore, the paper is not recommended for acceptance in its current form. I hope authors found the review comments informative and can improve their paper by addressing these carefully in future submissions.

**Additional Comments On Reviewer Discussion:**

There was no author response. During the discussion period, there was no other change. All reviewers stood by their original decision.

---

### Decision · Program_Chairs · 2025-01-22

Reject